# Fish Oil Nanoemulsion Supplementation Attenuates Bleomycin-Induced Pulmonary Fibrosis BALB/c Mice

**DOI:** 10.3390/nano12101683

**Published:** 2022-05-14

**Authors:** Danielle Galdino de Souza, Débora Silva Santos, Karina Smidt Simon, José Athayde Vasconcelos Morais, Luísa Coutinho Coelho, Thyago José Arruda Pacheco, Ricardo Bentes Azevedo, Anamélia Lorenzetti Bocca, César Augusto Melo-Silva, João Paulo Figueiró Longo

**Affiliations:** 1Nanobiotechnology Laboratory, Genetics & Morphology Department, Institute of Biological Science, University of Brasília, Brasília 70910-900, Brazil; danielle.galdino@hotmail.com (D.G.d.S.); debora.sqmc@live.com (D.S.S.); joseathayde_9@hotmail.com (J.A.V.M.); thyagojap@gmail.com (T.J.A.P.); razevedo@unb.br (R.B.A.); 2Applied Immunology Laboratory, Cell Biology Department, Institute of Biological Science, University of Brasília, Brasília 70910-900, Brazil; karina.smidt.simon@gmail.com (K.S.S.); luisa98@gmail.com (L.C.C.); albocca@unb.br (A.L.B.); 3Respiratory Physiology Laboratory, Faculty of Medicine, University of Brasília, Brasília 70910-900, Brazil; camelo@me.com

**Keywords:** nanoemulsion, lung fibrosis, bleomycin, fish oil

## Abstract

Diets rich in omega-3 or -6 fatty acids will produce different profiles for cell membranes phospholipid constitutions. Omegas 3 and 6 are part of the diet and can modulate the inflammatory profile. We evaluated the effects of the oral absorption of fish oil, when associated with a lipid nanoemulsion in an experimental pulmonary inflammatory model. Pulmonary fibrosis is a disease associated with excessive extracellular matrix deposition. We determined to investigate the morphophysiological mechanisms in mice that were pretreated after induction with bleomycin (BLM). The pretreatment was for 21 days with saline solution, sunflower oil (SO), fish oil (FO), and fish oil nanoemulsion (NEW3). The animals received a daily dose of 50 mg/Kg of docosahexaenoic acid DHA and 10 mg/Kg eicosapentaenoic (EPA) (100 mg/Kg), represented by a daily dose of 40 µL of NEW3. The blank group was treated with the same amount daily (40 µL) during the 21 days of pretreatment. The animals were treated with SO and FO, 100 mg/Kg (containing 58 mg/Kg of polyunsaturated fats/higher% linoleic acid) and 100 mg/Kg (50 mg/Kg of DHA and 10 mg/Kg EPA), respectively. A single dose of 5 mg/mL (50 μL) bleomycin sulfate, by the intratracheal surgical method in BALB/cAnNTac (BALB/c). NEW3 significantly reduced fibrotic progression, which can be evidenced by the protection from loss of body mass, increase in respiratory incursions per minute, decreased spacing of alveolar septa, decreased severity of fibrosis, and changes in the respiratory system. NEW3 attenuated the inflammatory changes developed in the experimental model of pulmonary fibrosis, while group SO showed a significant increase in inflammatory changes. This concluded that the presented results demonstrated that is possible to positively modulate the immune and inflamamtory response to an external agressor, by changing the nutitional intake of specific fatty acids, such as omega-3 placed in fish oil. Moreover, these benefits can be improved by the nanoencapsulation of fish oil in lipid nanoemulsions.

## 1. Introduction

Fibrogenesis is a natural physiological process characterized by extracellular matrix remodeling, typically observed in connective tissues [1]. This biological event is balanced by factors that promote extracellular matrix biosynthesis and tissue reabsorption, in order to provide tissue homeostasis [2].

In pulmonary connective tissues, this event is also present in normal conditions, but the disruption of this physiological condition can be abnormally triggered by external toxins, such as pollution, tobacco smoke, viral infection, or exposure to drugs, especially chemotherapeutical drugs, such as bleomycin (BLM) [3]. When fibrinogenesis is not followed by tissue reabsorption, the lung connective tissue tends to become scarred over time [2]. If this external trigger is continuous, it can produce an imbalanced situation, typically observed in chronic inflammation, and the consequence of this process is that respiratory function is irreversibly compromised [4].

The pathological condition, described previously, is called pulmonary fibrosis, and it is still poorly understood among researchers [5]. The main morphophysiological characteristics involve persistent damage to alveolar epithelial cells, subsequent release of pro-inflammatory and pro-fibrotic cytokines, accumulation of activated fibroblasts, myofibroblasts in fibrotic foci, excessive abnormal deposition of extracellular matrix proteins, reduction in the size of the lungs, and gradual loss of breathing capacity, contributing, consequently, to the increase in morbidity and mortality [6].

As commented previously, BLM is a drug associated with the development of pulmonary fibrosis. BLM has been used in cancer therapy, mainly in patients with Hodgkin’s Lymphoma, Kaposi’s Sarcoma, and cervical cancer; however, due to its fibrogenic properties, it has an adverse effect, and lung injury by BLM confers a mortality rate of 4–5% in these patients, demanding caution regarding its use [7].

Because of that, BLM has been extensively used as a pre-clinical animal model to study lung fibrosis pathogenesis and develop innovative therapeutic strategies to impair this pathological condition [8]. A single dose of BLM in rats by intratracheal instillate generates an irregular inflammatory and fibrotic process, starting with diffuse alveolitis and progressing with proliferated fibroblasts and extracellular matrix deposit [9].

Nowadays, commercially available therapeutic approaches to inhibit the development of pulmonary fibrosis are still limited, with only methods that slow down fibrotic progression [2,10]. There are currently two approved therapies: nintedanib and pirfenidone, with differentiated modes of action, but the same effectiveness in slowing the progression of the disease, despite not improving lung function [3].

Nintedanib (Ofev^®^) is an orally idiopathic pulmonary fibrosis drug that was approved for the first time in the U.S in 2014 [10]. This is a small, multiple receptor tyrosine kinase inhibitor developed by Boehringer Ingelheim with tolerability even in patients with co-morbidities. Pirfenidone, administered orally, has anti-inflammatory and antifibrotic properties with tolerability and gained approval only in 2014 in the U.S, while it has been approved since 2008 in Japan and 2011 in Europe [11].

Although the fibrotic process in the lung is still poorly understood among researchers, justifying intense studies regarding this pathology, it is suggestive that inflammatory modulation, when chronic and persistent, plays a key role in the intensification of pulmonary fibrosis [5].

Therefore, it is reasonable that strategies that could inhibit or attenuate the inflammatory process involved in pulmonary fibrosis could be applied to this pathology [12]. In the literature, nutritional strategies, aiming to modulate inflammation, have been published in the last few decades [13].

Following this idea, our research group recently published a report showing that oral supplementation with fish oil can be successfully used to control inflammation in some specific conditions. Furthermore, several reports in the literature have discussed the use of lipid nanoemulsions as a useful tool to improve the oral absorption of hydrophobic compounds [14,15]. Thus, the idea to combine fish oil with nanoemulsions is supported by the pharmacokinetic improvements provided by the nanocarriers [16].

For the present article, we aimed to evaluate this nutritional preventive supplementation with fish oil entrapped in lipid nanoemulsions in a chemically-induced pulmonary fibrosis animal model. One important novelty examined in this article was the morphophysiological pulmonary evaluation observed within the same group of experimental animals. Within this strategy, it was possible to correlate the morphological alterations with the functional and mechanical pulmonary physiology.

## 2. Materials and Methods

### 2.1. Methodological Steps

In order to investigate pulmonary morphophysiological changes, the study followed the following methodological steps: (1) pretreatment with saline solution (PBS) (healthy control), sunflower oil (SO), fish oil (FO), and fish oil nanoemulsion (NEW3); (2) inflammatory challenge with BLM for the development of pulmonary fibrosis (Figure 1).

### 2.2. Materials and Reagents

Omega-3 fish oil capsules (Omega-3 DHA-500; 1 g), containing a mixture of DHA and EPA (DHA 500 mg; EPA 100 mg) (obtained commercially from Naturalis Nutrição & Farma, Ltd., São Paulo, Brazil) and natural oil from babassu (BBS) extracted from palms (Cocais, Brazil) were tested in the oil phase nanoemulsion. Sunflower oil was commercially obtained by Cargill Agricola (Brasilia, Brazil) (composed 1.2 g of polyunsaturated fats/5% linoleic acid). Other reagents were needed for the experiment to proceed: phosphate buffered saline (PBS) (Interlab, Ltd., São Paulo, Brazil), Kolliphor^®^HS15 (Sigma-Aldrich, Darmstadt, Germany), and bleomycin sulfate (generously supplied by Applied Immunology Laboratory, University of Brasilia, Brasilia, Brazil).

### 2.3. Nanoformulation

The fish oil nanoemulsion (NEW3) used in the present report was prepared using a phase inversion temperature method, as previously described by our research group [17]. Fish oil samples were obtained from fish oil capsules (Naturalis Nutrição & Farma, Ltd., Curitiba, Brazil; 1 g) containing a mixture of DHA and EPA (DHA 500 mg; EPA 100 mg). DHA was used as a fish oil biomarker and measured to confirm the amount of DHA in the fish oil samples, as described by Santos et al. [16]. These results, as well as the nanoscopic characterization (mean hydrodynamic diameter of 156 nm; PDI of 0.34; and zeta potential of −16.9 mW), were confirmed by us in the present article.

For nanoemulsion preparation, oil phase, composed of fish oil (375 mg; containing 187.5 mg of DHA and 37.5 mg of EPA), BBS (0.1 g), and surfactant Kolliphor^®^HS15 (540 mg), were mixed, under stirring, for 15 min at room temperature (25 °C). The next step consisted of adding 10 mL of the aqueous phase (PBS), under continuous stirring. In this stage, a coarse emulsion was formed and heated to 85 °C. To form the stable nanoemulsion, 19.21 mL of cold water (4 °C) was rapidly added to the pre-emulsion. In this stage, a stable nanoemulsion oil in water nanoemulsion was formed. The fish oil nanoemulsion had an oil concentration of 12.5 mg/mL, which represents 6.25 mg/mL of DHA and 1.25 mg/mL of EPA.

### 2.4. Nanoemulsion Characterization

Nanoemulsion characterization was performed by measuring dynamic light scattering (DLS), which provides the hydrodynamic diameter. For the measurements, the nanoemulsions were dispersed (1:20) in Milli-Q ultrapure water (18 MΩ·cm) and measured using Zetasizer Nano (ZEN3690 model, Malvern Instruments, Malvern, UK), as described by using the same equipment, the electrophoretic mobility was measured to quantify the zeta potential (mV).

### 2.5. Animal Model

Female mice of the BALB/c strain (*Mus musculus*) (*n* = 8 per group) at 6 weeks of age, weighing between 20 and 25 g, were used. They were obtained in the central vivarium of the Catholic University of Brasília, Brazil, and kept in quarantine before the experiment was carried out in the vivarium of the Department of Genetics and Morphology, relocated on permanent ventilated shelves with wood shavings, controlled temperature (23 °C), light/dark cycle every 12 h, and receiving water and feed *ad libitum*. All procedures that were performed for anesthesia (ketamine [100 mg/kg], xylazine [10 mg/kg]), and euthanasia (propofol, 60 mg/kg, intravenously) of the animals were in accordance with the methods recommended by the Ethics Committee on Animal Use of the University of Brasília (N ° CEUA 97/2019).

### 2.6. Animal Treatments

Pretreatment was carried out for 21 days, using the gastric gavage method, with the experimental groups: (1) PBS, (2) SO, (3) FO, and (4) NEW3.

Pretreatment was carried out for 21 days, using the gastric gavage method, with the experimental groups: (1) PBS, (2) SO, (3) FO, and (4) NEW3. A total dose of 100 mg/kg, diluted in 200 µL, was used, considering the composition and dose studied by our research group [16].

During this pretreatment period, body weight was checked on days 0, 7, 14, and 21. The clinical signs of the animals were monitored every day.

After pretreatment, the groups (PBS, BLM, SO, FO, and NEW3) were submitted to the inflammatory challenge, first receiving the anesthetic solution composed of ketamine (100 mg/kg) and xylazine (10 mg/kg) (injected intraperitoneally) and, later, a single dose of 5 mg/mL (50 μL per animal) of bleomycin sulfate, by the intratracheal surgical method in mice, following the dose and procedure established in the study by [18]. The group of PBS animals received an equivalent volume of saline solution by the same route.

After inflammatory challenge, body weight was checked on days 0, 7, 14, 21, 28, and 35, and the respiratory frequency of the animals was monitored on the 29th day. The animals were euthanized 5 weeks (day 35) after the inflammatory challenge.

### 2.7. Respiratory Frequency

On the 29th day, after the inflammatory challenge, the animals’ respiratory rates (RR) were established, in order to analyze the number of respiratory incursions per minute (irpm) in approximately 60 s. The video recording method of each animal was used for 10 s; later, when it was able to measure the respiratory incursion for 10 s, multiply by six and obtain an approximate value of 1 min.

Afterwards, the video was submitted to the public domain application Efectum—slow, fast, and reverse camera (Craigpark Limited), using the slow tool to decrease the speed by 10 s, by 10× more, conditioning the duration of a new video acquired in approximately 60 s. The count was established by the observational method of inspiration per minute for each animal (60 s = total irpm).

### 2.8. Respiratory Mechanics Measurements

On the 36th day, after the inflammatory challenge, all animals were sedated and anesthetized with an anesthetic solution, composed of ketamine (100 mg/kg) and xylazine (10 mg/kg), injected intraperitoneally, for the insertion of an intratracheal cannula. They were mechanically ventilated with constant parameters in controlled volume mode: tidal volume of 8 mL/kg, with an ambient air and respiratory rate of 100 breaths/minute and positive end-expiratory pressure (PEEP) of 3 cmH_2_O.

Following the methodology of Melo-Silva et al. [19], the following steps were performed for the respiratory mechanics assessment: (1) performing the tracheostomy with an 18-gauge metal cannula; (2) administration of pancuronium bromide (0.1 mg/kg), intraperitoneally, to avoid respiratory efforts during respiratory system mechanics measurements; and (3) coupling to the computer-controlled FlexiVent FX mechanical ventilator (Scireq, Montreal, QC, Canada).

Before respiratory mechanics measurements, three deep inflations were performed, from PEEP to 30 cmH_2_O, to restore lung volume history. To measure respiratory system resistance (Rrs) and elastance (Ers), we applied a volume signal perturbation (containing one frequency) to the airway opening. Volume, airflow, and airway opening pressure signals were fitted to the respiratory system equation of motion [20]. In order to evaluate respiratory system impedance (Zrs), we applied a volume signal perturbation, containing frequencies between 0.25 and 19.62 Hz, to the airway opening. Zrs was fitted to the phase-constant model to obtain estimates of Newtonian resistance (R_N,_ central airway resistance), tissue damping (G, resistance of the small peripheral airways), and tissue elastance (H) [21]. We accepted models fit only if the coefficient of determination was >0.95. Up to five measurements were taken per animal, and the values were averaged.

### 2.9. Histopathological Examination

Sections of the right lung were established and stored in a 10% formaldehyde solution in saline buffer for fixing the material [18]. The material processing steps were followed, according to standard histological methods described by [22]. The samples were cut to a thickness of 5 µm in a Leica microtome model RM2125RT (Leica, Germany) and stained in the hematoxylin and eosin (H&E) methods (to identify inflammatory cells) and Masson’s Trichrome, in order to identify the collagen deposition, evidenced by the tone in blue (collagen fibers and nuclei stained blue and cytoplasm, muscle fibers and red blood cells stained with red). All samples of lung tissue affected by fibrosis, as well as sample controls, were stained and examined.

The slides containing samples from the lung sections, stained in H&E and Masson’s trichrome, were analyzed qualitatively and quantitatively and captured under the Invitrogen EVOS FL Auto Cell Imaging System (Thermo Fisher Scientific, Invitrogen, Carlsbad, CA, USA), with 10× magnification, 20×, 40× and 60×.

### 2.10. Alveolar Air Area

The H&E slides were quantified by digital image, in which five images of each animal were captured in a 20× objective and analyzed for the area of the pulmonary airspace. An average of 20 images per group was obtained, in order to try to approximate the reliability of the data obtained for pulmonary fibrosis [23].

The results of the pulmonary airspace area (white) were expressed as a percentage [24]. This involved defining the scale of the images (200 µm) to allow for the measurement of the distance from the airspaces or pulmonary parenchyma (depending on what is to be analyzed), converting them later into 8-bit (in gray-scale) and applying threshold to the images, using the black and white (B&W) tool to limit the intensity of the colors in the histopathological samples in front of the areas of interest (white tone for the airspace, including alveoli, alveolar, and bronchioles; black for the parenchyma pulmonary).

Thence, the results of the area measurements, expressed as a percentage, were obtained using the public domain software bundled with 64-bit, Windows version, ImageJ (version 1.51p) (National Institute of Health (NIH), Bethesda, MD, USA).

### 2.11. Ashcroft Score

The analysis of the histological classification of the degree of fibrous lesions was established by a blind semi-quantitative score, in view of the system adopted by Ashcroft et al. [23], related to the extent and severity of the inflammation and fibrosis in the lung parenchyma. Scores were divided into three categories: mild (0–3), moderate (4), and severe (5–8). The mean score of all sections was considered as the fibrosis score for that pulmonary section.

### 2.12. Computed Microtomography (Micro-CT)

The mice were scanned in the prone position on the 33rd day after the pulmonary inflammatory challenge. They were anesthetized using ketamine (100 mg/kg) and xylazine (10 mg/kg), injected intraperitoneally, to perform the procedure. The images were acquired in high resolution, with a voltage of 45 kV, 400 mA, 70 mm FOV (field of view measurement), totaling 1.000 projections, with a total scan time of 30 min. All images were acquired from a 360° angle, in the transversal, coronal, and sagittal planes, being reconstructed by the software Albira Suite 09-00127 Bruker (Ettlingen, Germany).

Analyses of two-dimensional (2D) reconstructions were performed using the commercial software PMOD—Biomedical Image Quantification Version 3.307 (Zurich, Switzerland). All acquisition and reconstruction parameters were the same for all mice. For the processing of images in three-dimensional (3D) reconstructions, the public domain software VolView Version 3.4 (Clifton Park, NY, USA) was used, an intuitive and interactive system for viewing the volume in 3D. However, before acquiring the images, automatic selections were made in the pulmonary area, in the transverse, coronal, and sagittal planes, thus allowing for images only of the organ of interest.

### 2.13. Statistics

All data are presented as mean ± standard deviation and analyzed statistically in the GraphPad Prism^®^ version 7.0 for Windows (GraphPad Software, San Diego, CA, USA, www.graphpad.com, accessed on 1 November 2019), subjected to specific statistical tests with 95% statistical confidence (*p* < 0.05). Statistical differences were assessed by the one-way ANOVA test, using Tukey’s multiple comparison tests.

## 3. Results and Discussion

In the last decades, there have been more studies in the literature indicating that nutrition is a significant variable that can impair or boost the immune system. This information has been used in medical care to modulate the immune system positively, avoiding several pathological diseases. As an example, fish oil has been extensively recommended as a nutritional supplement to improve the immunological response.

As a source of omega-3 polyunsaturated fatty acids (PUFAs), composed mainly of docosahexaenoic acid (DHA) and eicosapentaenoic (EPA), the positive effects of these compounds on the immune cell function were extensively investigated and are available in the literature [25].

For fish oil, supplementation is even more relevant because mammals are not able to synthetize these omega-3 PUFAs, thus nutrition is the only source able supply them [26]. Additionally, despite these immunological benefits, oral absorption of these constituents can be impacted, due to the low solubility of these compounds in the gastrointestinal tract.

Some research groups, including ours [14,16], have shown that the oral absorption of nanoemulsions can be significantly improved when it is associated with lipid nanocarriers. Furthermore, when oils are entrapped in lipid nanocarriers, the PUFAs constituents are chemically protected over time, increasing their stability for longer periods [16,27].

In other words, nanoencapsulation is used to extend natural oils shelf life and overcome some pharmacokinetics drawbacks, such as oral absorption. In this context, Figure 1 presents the fish oil nanoemulsion (NEW3) nanoscopic characterization. In Figure 2A, it is possible to observe the hydrodynamic diameter dispersion, with a mean value of 156 nm, while, in Figure 2B, the zeta potential dispersion is presented, with a mean value of −16.9 mV. The nanoscopic characterization is one important step of any kind of nanoemulsion. Understanding how the nanoemulsion is organized, in terms of size and electrophoretic mobility, can help us understand the biological behavior, as well as overtime stability [28,29].

### 3.1. Nanoformulation Attenuates Impacts on Body Weight

In the present article, we studied how previous supplementation with fish oil entrapped in lipid nanoemulsions could prevent or attenuate lung fibrosis induced by bleomycin, a common chemotherapeutic drug used in oncology. The first results are related to the body weight evaluation during the supplementation period (from day −21 to day 0, Figure 3A) and after BLM exposure (from day 1 to day 35, Figure 3B). In both evaluations, all experimental groups had similar body weight values at the end of the experimental observations. This is an important observation, since the nanoformulation did not promote an impact on the overall body weight, which may reflect normal food consumption during the experimental period.

The only exception was the group supplemented with SO, a positive control rich in omega-6 PUFAs (Figure 3B). This positive control was used because omega-6 fatty acids are well-known as pro-inflammatory nutrients. The loss of body weight of animals submitted to the inflammatory challenge of pulmonary fibrosis was evidenced in studies by Manali et al. [30], Chen et al. [31], and Zhao et al. [32], mainly between days 1 and 5, with a gradual increase in body weight from day 7.

### 3.2. Nanoformulation Attenuates Impacts on Respiratory Rate

Following the methodology strategy, we also measured some physiological and mechanical aspects of lung function that help us to understand the impact of nutritional supplementation. First, we measured animals’ respiratory rate, which is correlated with lung function. As presented in Figure 3C, the only statistical difference was observed in animals treated with sunflower oil, with a significant increase in respiratory rate, in comparison to the healthy animals. This significant increase in respiratory rate could be a response of the respiratory control centers to the impaired respiratory system mechanics, as expressed by augmented Rrs, Ers, R_N_, G, and H, as seen in SO treated animals.

### 3.3. Nanoformulation Attenuates Morphological Aspects of Pulmonary Fibrosis

Pulmonary fibrosis, induced by bleomycin, in mice remains unpredictable, in relation to its development, with variability in the extent of lung injury between the animals individually [33]. Regarding the micro-CT lung morphological aspects evaluated during the experiments, we collected some data that measured both qualitative and quantitative observations. For the analyses (Figure 4A), we noted that the animals treated with NEW3 presented lung images compatible with the normal aspect, animals treated with PBS, while animals treated with SO, and only with BLM, presented a morphological aspect with dispersed with spots, which can be compatible with the increase in tissue density in these animals.

Fibrosis is defined by the presence of radiological findings in high-resolution tomography: irregular linear opacity, parenchymal bands, traction bronchiectasis, and pulmonary distortion [34]. The anatomical distribution of these pulmonary changes corresponds to all lobes of the right and left lungs [35]. For the quantitative analysis in the micro-CT (Figure 4A), the data followed the qualitative observation, where the NEW3 animals presented a data dispersion displaced to the left, which is compatible with animals in healthy conditions. On the other hand, animals treated with SO and only BLM presented the data dispersion displaced to the right, indicating a higher lung tissue density in these animals.

Micro-CT analyses are an interesting approach for pre-clinical investigation, since this is a methodology that evaluates live animals, producing in vivo three-dimensional imaging analysis, and could be comparable to other physiological measurements, such as respiratory rate, as performed by us in the experimental design [36,37].

However, Micro-CT does not allow for differentiation between fibrosis and inflammation, and it is necessary to further explore this application in lung diseases [8]. Nevertheless, we showed in our study that micro-CT can offer a qualitative analysis of the effect of NEW3, related to pulmonary fibrosis, when an understanding of radiopacity and density is established, with it being possible to evidence the reduction in mice treated with the nanoformulation (Figure 4A).

In addition, as we are measuring values that could also be used for clinical studies, it is a methodology that has greater potential to be compared in translational technologies. Furthermore, in terms of ethical considerations, the use of live mice for evaluation is an important trend in life sciences experimentation, since it can reduce the number of experimental mice involved in the protocols.

Histopathological analysis is the standard method for the diagnosis and staging of pathological processes, but the development of fibrosis is distributed in a heterogeneous way by pulmonary extension, presenting areas that, for reasons not yet understood, are not affected [38].

Following the lung morphological analyses, we carried out some histopathological (Figure 4B) investigations, aiming to look for the potential structural alterations among the experimental groups. In addition, we also analyzed the amount of collagen-like tissues in the lung parenchyma, among the differential experimental groups. The morphological aspect of lung histology images showed that the healthy control group (PBS) presented a similar pattern to the animals treated with NEW3, while animals from the other experimental groups presented lung morphology that was compatible with lung fibrosis.

In addition, animals treated with SO and BLM presented the most prominent morphological alterations, presenting thickening of the alveolar wall, with an important reduction of the alveolar airspace.

These findings were also observed in the Masson (Figure 4B) staining analyses, when the collagen-like components are histochemically stained. This experiment is quite important, since it measures the amount of extracellular matrix produced by injured lungs after BLM exposure. The exacerbated production of collagen and other fibrotic proteins in pulmonary parenchyma form the biological processes that promote the reduction in lung elasticity and, consequently, the respiratory capacity of individuals affected by the disease.

This pulmonary hardening reflects the scarred tissue and can be seen in the histological images. In addition, the increase in the parenchyma tissue is graphically represented by the significant reduction in the alveolar airspaces. In Figure 4C, it is possible to observe that the alveolar airspace is significantly reduced in the animals treated with BLM, SO, and FO, while animals treated with NEW3 had alveolar airspace, similar to that in the healthy control animals (PBS). The reduction of the pulmonary air space occurs due to alveolar collapse, with air space obliteration after apposition of the alveolar walls, corresponding to one of the characteristics that occurred in fibrotic progression [39].

This pattern followed the Ashcroft analysis (Figure 4D), which measures the lung fibrosis intensity, in a semi-quantitative analysis. In summary, the morphological results showed that FO and NEW3 reduced lung fibrosis, with better results observed in the NEW3 treatments, while the SO treatment amplified lung fibrosis, in comparison to the control groups. Fish oil nanoencapsulation supports the results observed and can be explained by the better oral absorption of the oil, as showed in previous publications.

The theory around this is the dispersion of the fish oil molecules in the aqueous mucous layer provided by the nanoemulsion, which increases the contact among the FO components and mucosal cells layers. Furthermore, the nanodroplets keep a high surface area ratio, which enhances the contact between fatty acid molecules and the layers of the gastrointestinal mucosa. On the other hand, the fish oil administration alone tends to aggregate in bigger oil droplets when it comes into contact with the aqueous mucous layer. In this situation, the fish oil molecules are less likely to come into contact with the cells of the gastrointestinal mucosa, thus reducing its absorption.

### 3.4. Nanoformulation Attenuates Physiological Aspects of Pulmonary Fibrosis

The respiratory function measured is strongly correlated to lung mechanical properties, which are also connected to the tissue fibrosis detected in the previous morphological studies.

In addition, mechanics and respiratory function are affected by pulmonary fibrosis, and an analysis of the parameters is justifiable [40]. Regarding the respiratory system mechanics, we measured different experimental endpoints, aiming to understand the impact of fibrosis in respiratory mechanical properties.

These measurements were conducted *ex vivo* and included Rrs and Ers, as well as R_N_, G, and H (Figure 5A–E). For all these methodologies, we identified a similar pattern, with statistical difference for SO treatments, while NEW3-treated animals presented results similar to the PBS-treated mice.

It is important to review here that the physiological presence of collagen in the airways is a key point for analyzing the pulmonary mechanical properties associated with abnormal deposition [6]. Pulmonary mechanical changes may be associated with increased collagen fibers in the alveolar walls [9,41], as evidenced in Masson (Figure 4B).

Furthermore, studies performed with an experimental Balb/c animal model of pulmonary inflammation found a significant increase in airway resistance, viscoelastic pressure, static elastance, tissue damping, and tissue elastance, justified by the deposition of collagen fibers in the alveolar walls and fragmentation of elastic fibers [42]. These characteristics could also be observed in an experimental model of pulmonary fibrosis induced by bleomycin. In this study, changes in mechanical parameters and a decrease in respiratory function were observed in BALB/c mice [43].

The findings of the present study corresponded with clinical characteristics of human beings in the pathological condition of pulmonary fibrosis, corresponding to altered mechanical parameters of resistance and elastance, as evidenced by the significant increase [44], thus demonstrating the validation of the experimental animal model BALB/c submitted to inflammatory challenge with bleomycin and, in addition, serving as an analogue for research on lung disease.

### 3.5. Omega-3 Fatty Acids Promote Effects on Inflammatory Resolution

As commented previously, the largest differences observed in our experimental groups were observed among animals supplemented with SO and NEW3. These results can be explained by the PUFA constituents in the different oil sources. FO is rich in omega-3 PUFAs, while SO is mainly composed of omega-6 fatty acids.

This fatty acid constitution is determinant to create a more or less intense inflammatory environment [45]. The fatty acids are lipid components that can be used as an energy source or building blocks for cell structures, mainly for cell membrane phospholipids [26]. Due to this, cell membranes fatty acid constitution is strongly affected by the diet.

Diets rich in omega-3 or -6 fatty acids will produce different profiles for cell membranes phospholipid constitutions. The literature present strong evidence that omega-6-rich sources, such as sunflower oil, produce cellular environments more prone to inflammation, while omega-3-rich sources, such as fish oil, create tissues with a tendency to generate milder inflammatory processes [25].

One theory that explains this condition is that inflammation triggered by chemical aggression is usually initiated by cell membrane lysis, followed by fatty acids, released due to the activity of cellular and interstitial phospholipases. In this condition, fatty acids are released and used as substrate for constitutive cyclooxygenase-2 (COX-2) to produce the eicosanoid inflammatory mediators.

Moreover, the different sources of fatty acids, omega-3 or -6, compete in this enzymatic reaction, and the products for each reaction are different. Omega-3 fatty acids will produce prostaglandins and leukotrienes from the odd series (PGE-3, PGE-5, and LTB5). On the other hand, omega-6 fatty acids will generate these inflammatory mediators from the even series (PGE-2, PGE-4, and LTB4) [26]. Comparing the inflammatory activities of these two groups of mediators, the odd series produces less intense inflammation. Furthermore, the omega-3 fatty acids can produce resolvin, protectin, and maressin mediators that have strong anti-inflammatory activity and act to reduce the duration of inflammation [46].

Thus, the constitutive membrane fatty acid profile can determine the intensity, as well as the duration, of the inflammatory process. This is quite interesting, since it is not a medication strategy—it is totally based on nutrition quality. In other words, the conclusion of this report is that is possible to positively modulate the immune and inflammatory response to an external aggressor by changing the nutritional intake of specific fatty acids, such as omega-3 placed in fish oil. Moreover, we also can conclude that the nanoemulsion evaluated here can improve the effects of these fish oil fatty acids.

## Figures and Tables

**Figure 1 nanomaterials-12-01683-f001:**
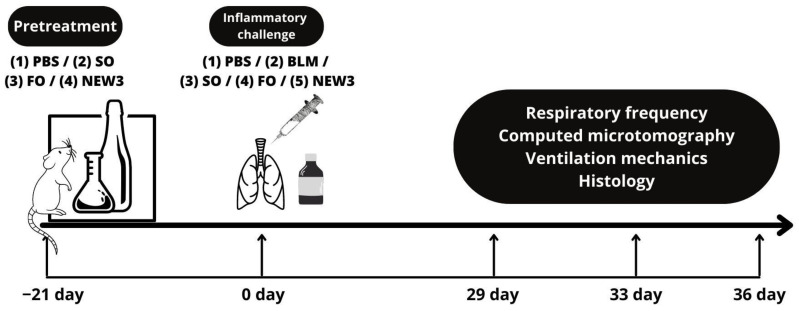
Time course of treatment for investigating pulmonary morphophysiological changes. Pretreatment: four groups of animals received intragastric gavage for 21 days, saline solution (PBS) (healthy control), sunflower oil (SO), fish oil (FO) components, and fish oil nanoemulsion (NEW3). Inflammatory challenge: after 21 days, the inflammatory challenge with bleomycin was performed. On days 29, 33, and 36, the procedures were established: respiratory frequency, computer microtomography, ventilation mechanics, and separate materials for histology.

**Figure 2 nanomaterials-12-01683-f002:**
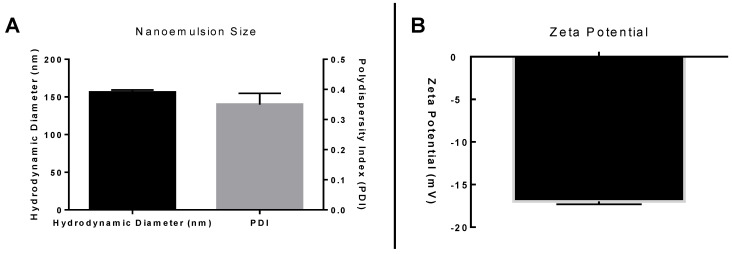
Fish oil nanoemulsion (NEW3) nanoscopic characterization. Section (**A**) shows the nanoemulsion hydrodynamic diameter (nm) and polydispersity index (PDI). Section (**B**) presents the Zeta potential (mW).

**Figure 3 nanomaterials-12-01683-f003:**
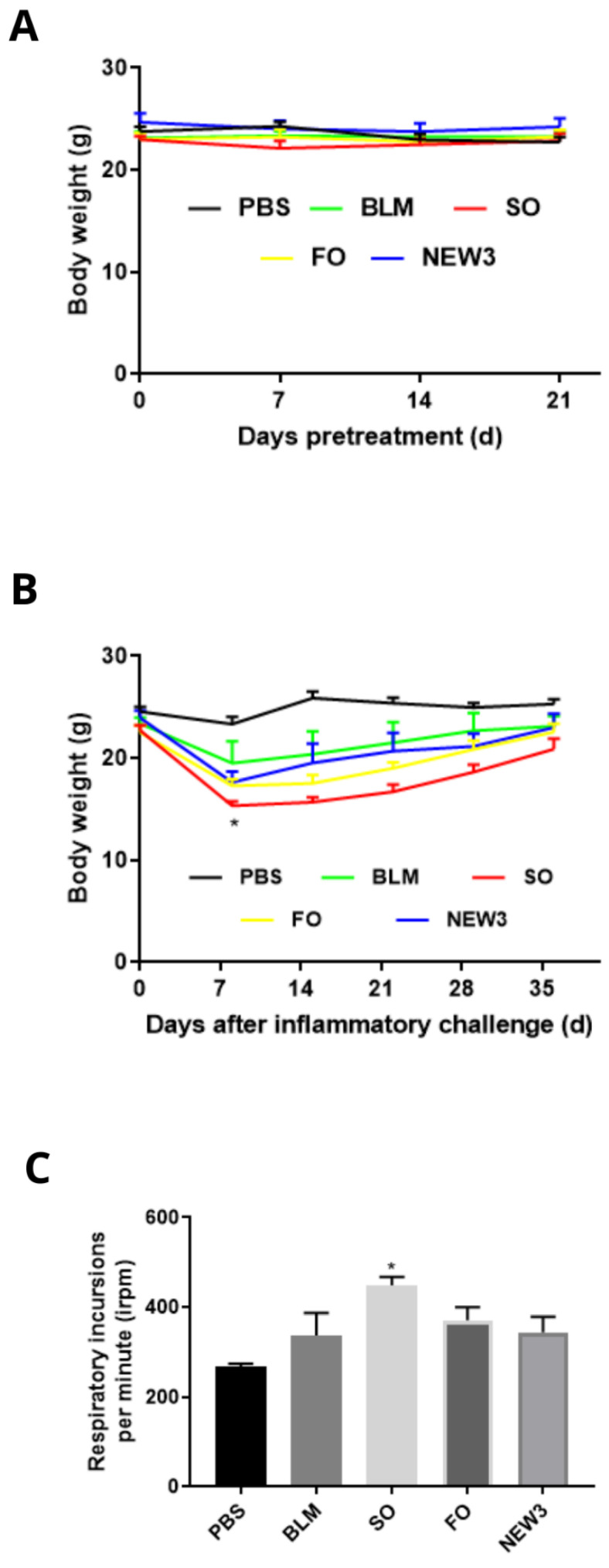
Pulmonary clinical aspects. (**A**) Body weight of mice during pre-treatment over 21 days. (*n* = 8). Legend: PBS: negative control group; BLM: positive control group; SO: group pretreated with sunflower oil (100 mg/kg); FO: animals pretreated with fish oil (100 mg/kg); NEW3: animals pretreated with fish oil nanoemulsion (100 mg/kg). (**B**) Body weight of mice over 36 days after pulmonary inflammatory challenge (*n* = 8). Legend: PBS: negative control group; BLM: positive control group challenged with BLM (5 mg/mL); SO: group pretreated with sunflower oil (100 mg/kg) and challenged with BLM (5 mg/mL); FO: group pretreated with fish oil (100 mg/kg) and challenged with BLM (5 mg/mL); NEW3: group pretreated with fish oil nanoemulsion (100 mg/kg) and challenged with BLM (5 mg/mL). Data are represented as mean ± standard deviation. Values were submitted to Tukey’s multiple comparison tests; * represents the statistically significant difference (*p* < 0.05), compared to the other experimental groups. (**C**) Respiratory rate on the 29th day after inflammatory challenge (*n* = 8). Caption: Irpm: respiratory incursions per minute. PBS: negative control group; BLM: positive control group challenged with BLM (5 mg/mL); SO: group pretreated with sunflower oil (100 mg/kg) and challenged with BLM (5 mg/mL); FO: group pretreated with fish oil (100 mg/kg) and challenged with BLM (5 mg/mL); NEW3: group pretreated with fish oil nanoemulsion (100 mg/kg) and challenged with BLM (5 mg/mL). Data are represented as mean ± standard deviation. Values were submitted to Tukey’s multiple comparison tests; * represents the statistically significant difference (*p* < 0.05), compared to the PBS group.

**Figure 4 nanomaterials-12-01683-f004:**
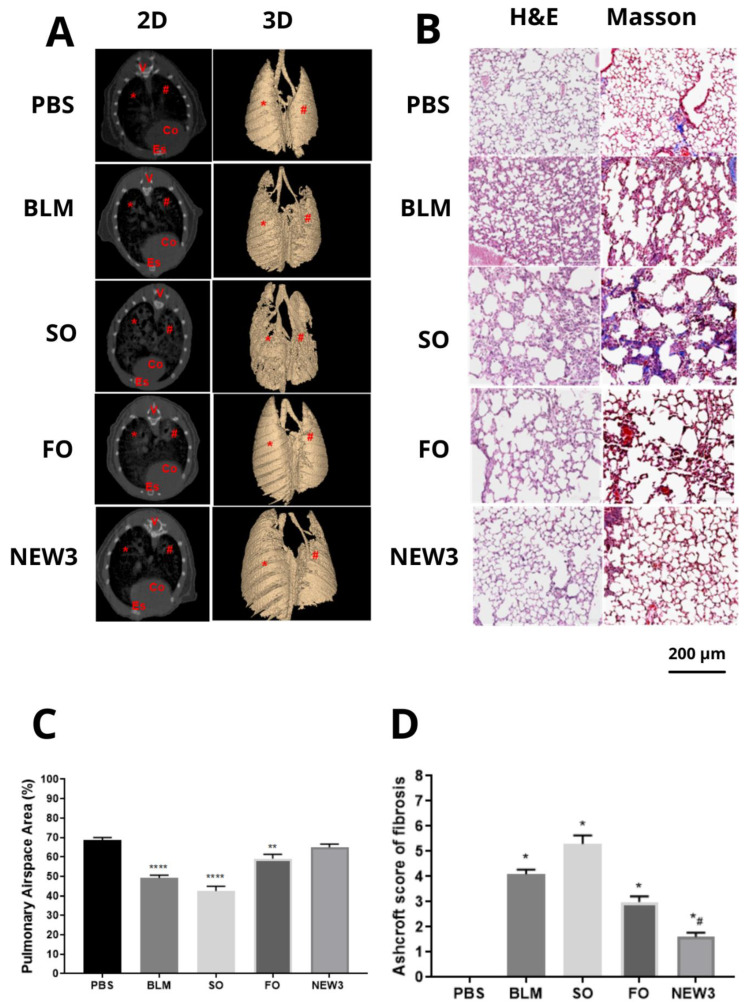
Pulmonary morphological aspects. (**A**) Micro-CT imaging after an inflammatory challenge. PBS, BLM, SO, FO, and NEW3 cross-section represented in 2D, and anterior region of the lungs represented in 3D of an animal; Co: heart; V: vertebra; Es: sternum bone. * right lung; # left lung. (**B**) Histology of the pulmonary parenchyma of BALB/c mice after an inflammatory challenge, stained using H&E and Masson’s trichrome techniques. PBS, control group treated with saline solution, seen by the H&E and Masson. BLM, positive control group treated with bleomycin, as seen by the H&E and Masson technique. SO treated with sunflower oil, as seen by the H&E and Masson. FO treated with fish oil, as seen by the H&E and Masson. NEW3 treated with fish oil nanoemulsion, seen by the H&E and Masson; 200 µm size reference bar. (**C**) Pulmonary airspace area. PBS: negative control group; BLM: positive control group, challenged with BLM (5 mg/mL); SO: group pretreated with sunflower oil (100 mg/kg) and challenged with BLM (5 mg/mL); FO: group pre-treated with fish oil (100 mg/kg) and challenged with BLM (5 mg/mL); NEW3: group pretreated with fish oil nanoemulsion (100 mg/kg) and challenged with BLM (5 mg/mL). Images (total: 20 for each group) of the lung parenchyma were captured to assess the air spacing of each group. The data are represented according to the mean ± standard deviation. The values were subjected to Tukey’s multiple comparison tests; ** represents the statistically significant difference (*p* < 0.05), compared to the PBS group; **** represents the statistically significant difference (*p* < 0.05), compared to the PBS group. (**D**) Scale of the degree of pulmonary fibrosis. Legend: PBS: negative control group; BLM: group positive control, challenged with BLM (5 mg/mL); SO: group pretreated with sunflower oil (100 mg/kg) and challenged with BLM (5 mg/mL); FO: group pre-treated with fish oil (100 mg/kg) and challenged with BLM (5 mg/mL); NEW3: group pretreated with fish oil nanoemulsion (100 mg/kg) and challenged with BLM (5 mg/mL). Data are represented according to the mean ± standard deviation. The values were subjected to Tukey’s multiple comparison tests; * represents the statistically significant difference (*p* < 0.05), compared to the PBS group; # represents the statistically significant difference (*p* < 0.05), compared to groups challenged with BLM.

**Figure 5 nanomaterials-12-01683-f005:**
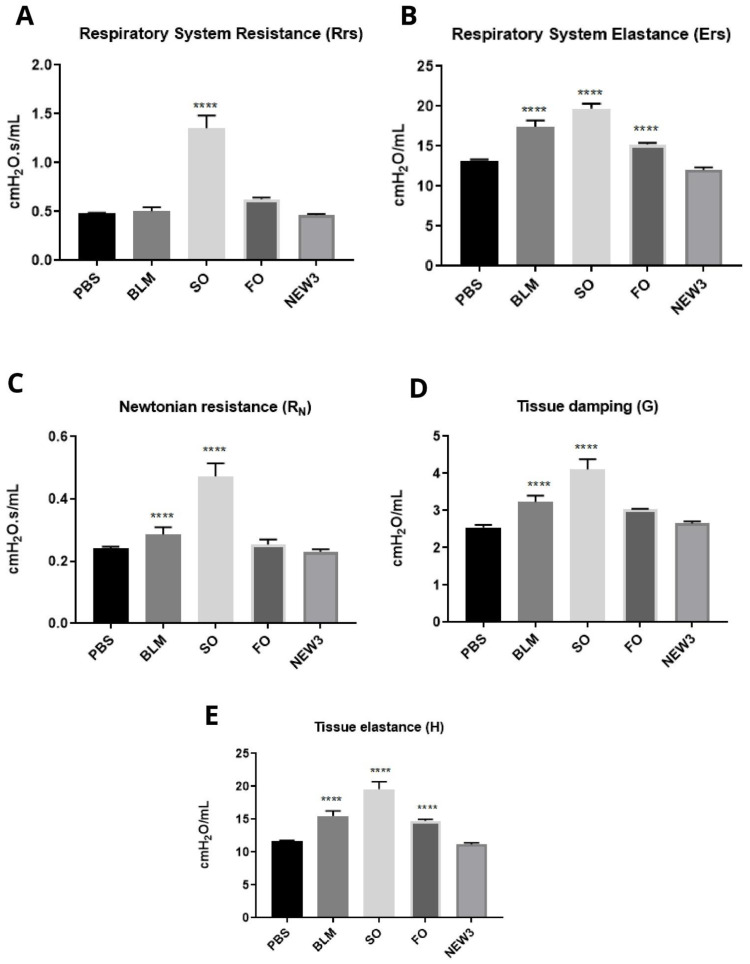
Pulmonary physiological aspects. (**A**) Respiratory system resistance (Rrs) on day 36, after inflammatory challenge (*n* = 8). (**B**) Respiratory system elastance (Ers) on day 36, after inflammatory challenge (*n* = 8). (**C**) Newtonian resistance (R_N_) on day 36, after inflammatory challenge (*n* = 8). (**D**) Tissue damping (G) on the 36th day, after inflammatory challenge (*n* = 8). (**E**) Tissue elastance (H) on day 36, after inflammatory challenge (*n* = 8). Legend: PBS: negative control group; BLM: positive control group, challenged with BLM (5 mg/mL); SO: group pretreated with sunflower oil (100 mg/kg) and challenged with BLM (5 mg/mL); FO: group pretreated with fish oil (100 mg/kg) and challenged with BLM (5 mg/mL); NEW3: group pretreated with fish oil nanoemulsion (100 mg/kg) and challenged with BLM (5 mg/mL). Data are represented as mean ± standard deviation. Values were submitted to Tukey’s multiple comparison tests; **** represents the statistically significant difference (*p* < 0.05), compared to the PBS group.

## Data Availability

Not applicable.

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
