# Peer review of "Fish Oil Nanoemulsion Supplementation Attenuates Bleomycin-Induced Pulmonary Fibrosis BALB/c Mice"

_nanomaterials, 2022, doi:10.3390/nano12101683_

Round 1
Reviewer 1 Report
The authors investigated the use of a fish oil nanoemulsion to treat bleomycin-induced fibrosis. This is a fascinating study, and I applaud the author's efforts. However, the methods and figures in the current version of the manuscript need to be significantly improved.
Comments
The method of nanoemulsion preparation was described in general, and it should be more specific to the fish-oil -nanoemulsions. The authors should include respective figures, characterization methods of the fish oil nanoemulsion preparation, and characterization in the results and discussion section.
The section on animal model development and nanoemulsion administration requires significant improvement.
Author Response
Reviewer 1:
Comments and Suggestions for Authors
The authors investigated the use of a fish oil nanoemulsion to treat bleomycin-induced fibrosis. This is a fascinating study, and I applaud the author's efforts. However, the methods and figures in the current version of the manuscript need to be significantly improved.
Comments:
The method of nanoemulsion preparation was described in general, and it should be more specific to the fish-oil -nanoemulsions. The authors should include respective figures, characterization methods of the fish oil nanoemulsion preparation, and characterization in the results and discussion section.
The section on animal model development and nanoemulsion administration requires significant improvement.
Authors Response:
First of all, on behalf of the authors, I would like to thank the reviewers’ words. Regarding the suggested points, we will present a rational point-by-point revision:
- The nanoemulsion characterization methodology was revised. We included the following sentence in the revised manuscript:
Among lines 136-142:
2.4. Nanoemulsion Characterization
Nanoemulsion characterization was performed by measuring the dynamic light scattering (DLS), which provides the hydrodynamic diameter. For the measurements, the nanoemulsions were dispersed (1:20) in Milli-Q ultrapure water (18 MΩ.cm) and measured using Zetasizer Nano (ZEN3690 model, Malvern Instruments, United Kingdom), as described by. Using the same equipment, the electrophoretic mobility was measured to quantify the Zeta Potential (mV).
Additionally, we include the following sentences in the Results and Discussion section (292-300):
“In other words, nanoencapsulation is used to extend natural oils shelf life, and to overcome some pharmacokinetics drawbacks, such as oral absorption. In this context, Figure 1 presents the fish oil nanoemulsion (NEW3) nanoscopic characterization. In Figure 1A it is possible to observe the hydrodynamic diameter dispersion with a mean value of 156 nm; while in Figure 1B the Zeta Potential dispersion is presented with a mean value of -16.9 mV. The nanoscopic characterization is one important step of any kind of nanoemulsion. Understanding how the nanoemulsion is organized in terms of size and electrophoretic mobility can help us understand the biological behavior, as well as overtime stability [28,29].”

Reviewer 2 Report
The research article evaluates the nutritional preventive supplementation with fish oil entrapped in lipid nanoemulsions in a chemically induced pulmonary fibrosis animal model.
This is a well-presented original study, with a sound hypothesis, but certain incorporation of methodological studies to be included for proper presentation of the results.
The manuscript can be considered for publication once the below mentioned points are addressed.
- The prepared fish oil nanoemulsion needs to be characterized and added to the current manuscript. Characterization studies are required to evaluate various physicochemical parameters of the NEW3 like morphology (SEM, TEM), refractive index, particle size and their distribution, zeta potential, percentage transmittance, dilution test, emulsifying time, dye test, drug content, viscosity and drug release.
The following papers can be referred:
Development and characterization of nanoemulsion as carrier for the enhancement of bioavailability of artemether (DOI: 10.3109/21691401.2014.887018)
Nanoemulsions: Formulation, characterization, biological fate, and potential role against COVID-19 and other viral outbreaks (DOI: 10.1016/j.colcom.2021.100533)
Nanoemulsion preparation, characterization, and application in the field of biomedicine (DOI: 10.1016/B978-0-12-816200-2.00019-0)
- The drug release studies for the NEW3 are to be included. In vitro and ex vivo drug release needs to be studied to interpret the mechanism of drug release
- Why female Balb/c mice were used for this study? Kindly justify with proper explanation and reference in discussion.
- Cell Culture studies could have been performed to evaluate the anti-fibrotic potential of NEW3 on different cell lines.
- Line 440 – 451: Format and line spacing to be checked
- Certain grammatical errors are present in the manuscript which needs to corrected.
Author Response
Reviewer 2:
Comments and Suggestions for Authors
The research article evaluates the nutritional preventive supplementation with fish oil entrapped in lipid nanoemulsions in a chemically induced pulmonary fibrosis animal model. This is a well-presented original study, with a sound hypothesis, but certain incorporation of methodological studies to be included for proper presentation of the results. The manuscript can be considered for publication once the below mentioned points are addressed.
- The prepared fish oil nanoemulsion needs to be characterized and added to the current manuscript. Characterization studies are required to evaluate various physicochemical parameters of the NEW3 like morphology (SEM, TEM), refractive index, particle size and their distribution, zeta potential, percentage transmittance, dilution test, emulsifying time, dye test, drug content, viscosity and drug release.
The following papers can be referred:
Development and characterization of nanoemulsion as carrier for the enhancement of bioavailability of artemether (DOI: 10.3109/21691401.2014.887018)
Nanoemulsions: Formulation, characterization, biological fate, and potential role against COVID-19 and other viral outbreaks (DOI: 10.1016/j.colcom.2021.100533)
Nanoemulsion preparation, characterization, and application in the field of biomedicine (DOI: 10.1016/B978-0-12-816200-2.00019-0)
Authors Response:
Thanks for your suggestion. We included a new methodology section with the nanoemulsion nanoscopic characterization (Lines 136-142). Part of this characterization has been previously published by our group with the same nanoemulsion, however, we also understand that makes sense to include some technical information in the present article. The methodology section was included among lines 136-142 in the revised manuscript. A new revised Figure 1 was included in the revised version; thus, all the following figures were also updated. The description, as well as the discussion of the revised results, were placed among lines 292-300 in the revised manuscript. The suggested references were also cited in this last discussion.
- The drug release studies for the NEW3 are to be included. For in vitro and ex vivo drugs, the release needs to be studied to interpret the mechanism of drug release
Authors response:
Thanks for this suggestion. This manuscript is a follow-up to a previous paper published by our group using the same Fish Oil Nanoemulsion (Santos et al. 2021). In this paper, we investigated DHA absorption by cell cultures, and we also measured the in vivo nanoemulsion biodistribution, by using in vivo fluorescent measurements. For the present article, we focused on the in vivo biological application of the fish oil nanoemulsion, and how It can impact the prevention of lung fibrosis, as presented in the results section.
- Why female Balb/c mice were used for this study? Kindly justify with proper explanation and reference in the discussion.
Authors response:
In general female mice are less aggressive in comparison to male mice. In terms of aggressiveness, male mice are more susceptible to becoming aggressive under diverse experimental conditions, while female mice can be aggressive in more specific conditions, such as pregnancy or during post-partum periods. Based on this information, and as we aimed to conduct experiments in mice after bleomycin exposition, we design the study using female mice aiming at the reduction of potential aggressiveness among the experimental animals.
NEWMAN, Emily L. et al. Fighting females: neural and behavioral consequences of social defeat stress in female mice. Biological psychiatry, v. 86, n. 9, p. 657-668, 2019.
- Cell Culture studies could have been performed to evaluate the anti-fibrotic potential of NEW3 on different cell lines.
Authors response:
It is an interesting suggestion, however, lung fibrosis is a multi-complex biological event that involves several tissues and cell lines, including lung parenchyma and stroma, as well as the local immune system. To our knowledge, there is no in vitro system that can reproduce the biological behavior of this event. Additionally, we used a specific injury challenger, bleomycin, which is widely used as a chemotherapeutical drug. Thus, we understand that for our objectives, the in vivo study is still the best option for this kind of investigation.
- Line 440 – 451: Format and line spacing to be checked
Authors response:
Thanks for that observation. As a nonmandatory section, we will remove the conclusion section from the final revised version. The conclusions were placed in the last part of the results and discussion section. We also include a conclusion among lines 467-471: “In other words, the conclusion of this report, is that is possible to positively modulate the immune and inflammatory response to an external aggressor, by changing the nutritional intake of specific fatty acids, such as omega-3 placed in fish oil. Moreover, we also can conclude that the nanoemulsion evaluated here can improve the effects of these fish-oil fatty acids.”
- Certain grammatical errors are present in the manuscript which needs to corrected.
Authors Answer:
Thanks for this observation. We will revise the revised manuscript.

Reviewer 3 Report
The manuscript “Fish oil nanoemulsion supplementation attenuates bleomycin-induced pulmonary fibrosis BALB/c mice” evaluated the effects of fish oil formulated as a nanoemulsion in experimental pulmonary inflammatory mode. The data presented here substantiate the author's hypothesis and could be useful for treating or preventing prevent lung fibrosis. The study was well designed and executed in a good way. There are a few minor editing/corrections required to improve the quality of this article.
Comments
- A conclusion is missing in abstract section
- There is still scope for improvement in introduction, the number of references are relatively low.
- A clear difference from the earlier studies is missing in the introduction section.
- No characterization of the formulations?
- Include the images of the product in the results.
- Figure 2 images are not very clear. Please recheck the statistical significance in Figure 2b.
- A list of abbreviations could be useful for the readers.
Author Response
Reviewer 3:
Comments and Suggestions for Authors
The manuscript “Fish oil nanoemulsion supplementation attenuates bleomycin-induced pulmonary fibrosis BALB/c mice” evaluated the effects of fish oil formulated as a nanoemulsion in experimental pulmonary inflammatory mode. The data presented here substantiate the author's hypothesis and could be useful for treating or preventing prevent lung fibrosis. The study was well designed and executed in a good way. There are a few minor editing/corrections required to improve the quality of this article.
Comments:
A conclusion is missing in abstract section. There is still scope for improvement in introduction, the number of references is relatively low. A clear difference from the earlier studies is missing in the introduction section. No characterization of the formulations? Include the images of the product in the results. Figure 2 images are not very clear. Please recheck the statistical significance in Figure 2b.
A list of abbreviations could be useful for the readers.
Authors response:
Thanks for all your suggestions. Regarding the revision points:
Abstract conclusion: We included the following sentences for the abstract conclusion (Lines 31-34): “Concluding, the resuls presented demonstrated that is possible to positively modulate the immune and inflamamtory response to an external agressor, by changin the nutitional intake of specific fatty acids, such as omega-3 placed in fish oil. Moreover, this benefits can be improved by the nanoencapsulation of fish oil in lipid nanoemulsions.”
Introduction Improvement: we included the following sentences in the revised introduction section (Lines 91-94): “several reports in the literature have discussed the use of lipid nanoemulsions as a useful tool to improve the oral absorption of hydrophobic compounds [1, 2]. Thus, the idea to combine fish oil with nanoemulsions is supported by these pharmacokinetic improvements provided by the nanocarriers [14].”
Nanoemulsion characterization: we included a nanoscopic characterization, as well a new Figure 1 with the results obtained in this topic. The methodology and the results and discussion sections were included among lines 136-142 and 292-300 respectively.
Figure 2 images are not very clear. Please recheck the statistical significance in Figure 2b:
Authors Answer: the statistical analysis is correct. The only point that we found significance is for the SO group, at day 7.

Round 2
Reviewer 2 Report
The authors have revised the manuscript entitled "Fish oil nanoemulsion supplementation attenuates bleomycin-induced pulmonary fibrosis BALB/c mice" as per the reviewers comments. The revised version of the manuscript enhanced the clarity of review performed and is suitable for publication.